# Potential Role of Protocatechuic Acid as Natural Feed Additives in Farm Animal Production

**DOI:** 10.3390/ani12060741

**Published:** 2022-03-16

**Authors:** Shad Mahfuz, Hong-Seok Mun, Muhammad Ammar Dilawar, Keiven Mark B. Ampode, Chul-Ju Yang

**Affiliations:** 1Animal Nutrition and Feed Science Laboratory, Department of Animal Science and Technology, Sunchon National University, Suncheon 57922, Korea; shadmahfuz@yahoo.com (S.M.); mhs88828@nate.com (H.-S.M.); ammar_dilawar@yahoo.com (M.A.D.); keivenmarkampode@sksu.edu.ph (K.M.B.A.); 2Department of Animal Nutrition, Sylhet Agricultural University, Sylhet 3100, Bangladesh; 3Department of Multimedia Engineering, Sunchon National University, Suncheon 57922, Korea; 4Interdisciplinary Program in IT-Bio Convergence System (BK21 Plus), Sunchon National University, 255 Jungangno, Suncheon 57922, Korea

**Keywords:** protocatechuic acid, antioxidant function, immune function, anti-microbial properties, intestinal health, animals

## Abstract

**Simple Summary:**

Protocatechuic acid (PCA) is a phenolic-rich compound that exists in natural plants. Many in vitro studies have reported its antioxidant, anti-inflammatory, anti-microbial properties. Although the health-promoting effects of PCA have been established in human medicine, the applications of PCA as natural feed additives on farm animal production still need to be re-investigated. Therefore, the aim of the review study was to gather research information on PCA to better our understanding of the practical uses of those natural antioxidant-rich feed additives in animal fodder.

**Abstract:**

Restriction on using antibiotics in animal feed that generates demand for antibiotics alternatives in animal breeding. Providing safe food to humans free from the residual effects of antibiotics is a great challenge to animal producers and food-producing industry owners. Medicinal plants and their extracts as feed supplements have been used to promote the growth and health of farm animals for centuries. Protocatechuic acid (PCA) is a phenolic compound that originated from natural plants. For years, the health-promoting role of PCA has been becoming an attraction of research in nutrition and pharmacy. Thus, it can be used as an active natural feed additive while synthetic antibiotics are illegal to use in animal breeding. However, the practical application of PCA in view of dosages in animal nutrition, together with its mode of action on animal health, is not well known. In this regard, this review study has explored the mode of action of PCA and the feasibility of using those compounds in animal nutrition. This review study concludes that phenolic-rich protocatechuic acid as a natural feed additive may be useful in enhancing antioxidant status, immune function, antimicrobial, intestinal health and growth performance of farm animals.

## 1. Introduction

Protocatechuic acid (PCA) is a phenolic-rich compound that is derived from herbs. In recent years, numerous research findings reported that PCA, the major metabolite of complex polyphenols, was very effective in various physiological functions and health due to its anti-inflammatory properties [1,2,3]; antioxidant, peroxidation inhibition with free radical scavenging activities [4,5]; antibacterial activities [6]; antiviral activities [7]; anticancer activities, antiulcer activities, anti-fibrotic activities [8]; cardiac activities, hepatoprotective activities, neurological and nephron-protective activities [9,10,11]. An in vivo study confirmed that PCA has antioxidant activities by decreasing the oxidative stress in type 1 diabetes mellitus (T1DM) rats and by promoting the antioxidant enzymatic function which subdued the reactive oxygen species (ROS) generation [12]. PCA has been examined about its biological and therapeutic activities in various diseases model. In addition, PCA had a protective role against inflammation, inducing stress in human dermal fibroblast by ROS scavenging activities [2]. PCA has a role in immune function via producing immunoglobin and secretion of the anti-inflammatory cytokine, increasing phagocytosis, and suppressing pro-inflammatory cytokines [13]. PCA showed the antibacterial properties in a beef sample by hindering bacterial growth via decreasing lipid oxidation along with enhancing the synergistic role of antibiotics to reduce the drug resistance [14]. Besides, PCA has specific nutritional attention. It is a product of the anthocyanin metabolite that could influence tissues and has biological health effects [15]. Dietary supplementation of PCA was reported to have a greater deposition of PCA in blood and tissues sample (brain, heart, liver, and kidney) of mice for 12 weeks of study [16]. In addition, Tan et al. [3] reported that the PCA at 800 mg/L was very effective in improving the growth of freshwater microalgae (*Euglena gracilis*) and the accumulation of metabolites, namely chlorophyll, carotenoids, paramylon, etc.

Although PCA has many health-promoting effects, the utilization of PCA in farm animal production is still limited. Since PCA possesses antioxidant, anti-inflammatory, and anti-microbial properties which is the main goal to utilize those compounds as natural feed additives in farm animal production. There is still a knowledge gap on the effect of PCA on the intestinal mucosal barrier, intestinal tight junction protein expression, intestinal immunity, anti-inflammatory on supporting the intestinal health and the production performance of animals. Instead, restrictions on using antibiotics in feed create a demand for alternatives to antibiotics in the feed industry. Animal researchers are now spotting phenolic-rich compounds as effective feed additives for livestock farming [17]. The greater consciousness of the food quality, health safety issue, and the prohibition of antibiotic growth promoters in animal diets have reached the attention of food producers as well as industry owners regarding natural feed additives [18]. Consequently, this review study concentrates on the possibility of using PCA as natural feed additives in animal nutrition.

## 2. General Characteristics and Sources of Protocatechuic Acid

Protocatechuic acid (PCA) is chemically known as 3,4-dihydroxybenzoic acid. It has an aromatic ring with one or more hydroxyl groups. PCA exhibits a common structure with some reported antioxidants like gallic acid, caffeic acid, vanillic acid, and syringic acid, etc. [10]. It is a dark gray solid crystalline powder, having a melting point of 221 °C, and a boiling point of 410 °C with 760 mm Hg [10,19]. PCA has a phenolic odor. It is soluble in water, alcohol, ether and fades in the air [10]. PCA compounds that arise from a direct or indirect source of secondary metabolites of different plants polyphenols, for example, anthocyanins, proanthocyanidins, phenylalanine, quercitins, etc. It is an extensively dispersed phenolic-rich compound found in different herbs, grains, and fruits including plums, grapes, nuts (almonds), onions (*Allium cepa* L.), grain brown rice (*Oryza sativa* L.), and olives (*Olea europaea*), roselle (*Hibiscus sabdariffa*), calamondin (*Microcarpa bunge*), white wine grapes (*Vitis vinifera*) and citrus fruits [2,20]. It is a major metabolite of anthocyanin and anthocyanins which is reported to be effective in reducing the risk of cardiovascular diseases in humans [21]. Those existing secondary metabolites can transform into PCA through different bioactive processes and possess effective antioxidant properties. For instance, PCA was found in the plasma of mouse fed cyanidin at 2 g/kg diets for 14 days and was effective to protect ROS [22]. The authors claimed that PCA was derived from the degradation of cyanidin products. It was also reported that in acidic conditions, the anthocyanin became stable, but it turns into PCA at a neutral pH [22]. Furthermore, the intestinal flora may have a role to transform secondary metabolic into PCA [23]. The common plant sources of PCA are presented in Table 1 and the chemical structure is shown in Figure 1.

## 3. Biological Role of Protocatechuic Acid

### 3.1. Role of Protocatechuic Acid on Antioxidant Enzyme Activities

Free radicals namely reactive nitrogen species (RNS, nitric oxide) and reactive oxygen species (ROS, superoxide, hydrogen peroxide, hydroxyl) are normally produced during aerobic metabolism. However, body tissues cannot usually neutralize those excessively produced pro-oxidant molecules during any stress. As a result, those free molecules cause severe injuries in chromosomes and other body processes [18,43]. Oxidative stress causes great economic losses to livestock farming. Antioxidants are known to be useful in preventing diseases caused by oxidative stress. Thus, to save the biomolecules from ROS and RNS, different synthetic antioxidants have been applied in the food processing industry. However, those synthetic antioxidants have been reported as harmful to body tissues while plant-based antioxidants have been recognized as better than synthetic antioxidants due to their lower cytotoxicity and residual effects [44]. Thus, considerable attention has been paid to searching for alternative synthetic antioxidants from natural sources or plant-based origins. Therefore, the establishment of PCA as natural antioxidant enrich feed additives may solve the above issue.

In different biochemical assays, PCA at the rate of 0.05 to 0.10 mg/mL could increase the inhibition of lipid peroxidation by thiobarbituric acid reactive substances (TBARS) assay, scavenging of hydrogen peroxide (H_2_O_2_), and diphenyl picrylhydrazyl (DPPH) by a corresponding assay which ensured its antioxidant characteristics [45]. The antioxidant role of PCA was noted by the stimulation of mRNA transcription of glutathione (GSH) related enzymes, decreasing the apoptosis rate by ROS reduction, enhancing the mitochondrial function, and suppression of DNA fragmentation in vitro (J774 A.1 macrophages and Human neuronal cell line) [46,47,48]. PCA could protect rats from oxidative stress persuaded by tert-butyl hydro-peroxide (t-BHP); pretreatment with PCA could reduce the serum lactate dehydrogenase (LDH) levels, alanine aminotransferase (ALT), aspartate aminotransferase (AST); and could protect liver tissue from the oxidative stress [10]. Furthermore, PCA compounds have been reported for their metal chelating properties namely iron and copper, that ultimately protect to form free radicals from metal-catalyzed molecules [49]. Farombi et al. [50] reported that the antioxidant capacity of PCA was ten times more than that of Vitamin E. An in vitro study was done using different assays where trolox was considered as a standard antioxidant and the results showed that PCA had higher antioxidant activities through chelating metal transition ions and by scavenging the free radicals through donating hydrogen or electron in lipid and aqueous media. The authors recommend that PCA can be used as a natural source of antioxidants to improve food quality and preservation in the food industry [51]. Furthermore, the intestinal mucosal surface is usually attacked by ROS, preventing the absorption of nutrients, while an antioxidant could neutralize ROS and could protect the intestinal surface from injuries [52]. Song et al. [19] noted that the basic mode of action for antioxidant properties of PCA, may be due to the combination of reducing the level of anti-inflammatory markers, up-regulating the endogenous antioxidant enzyme activities, regulating different signaling pathways, and preventing oxidative damage. However, the research on the antioxidant properties of PCA in farm animals is still very limited.

### 3.2. Antimicrobial Properties of Protocatechuic Acid

The antimicrobial properties of PCA originating from different plants have been defined in various reports [53,54]. For instance, the antibacterial role of PCA derived from Roselle calyx (*Hibiscus sabdariffa* L.) extract was found in ground beef tissue while in storage at 4 °C for 15 days [32]. The meat quality parameters including cooking loss, pH, fat%, protein% and moisture% were not affected due to the inclusion of PCA in beef samples. The authors concluded that PCA can be used as a food preservative to prevent spoilage from organisms. In another study, PCA at 5 mg or 10 mg/100 g was added into a ground beef sample that ensured the antimicrobial activities of PCA against *Campylobacter* spp. [14]. Moreover, PCA at 2000 μg/mL could subdue the growth of *Pseudomonas aeruginosa* and could increase the synergistic effects of antibiotics to reduce drug resistance [55]. Kuete et al. [56] also reported that PCA at 1.22–625 μg/mL could prevent about 80% growth rate for the bacteria and fungi. In an in vitro study, PCA at 8–64 mg/L could inhibit the growth of *Helicobacter pylori* in a beef sample [57]. The antimicrobial role of PCA might be related to decreasing the lipid oxidation in beef tissue. It was hypothesized that the presence of PCA-rich phenolic compounds at the cellular level results in bacterial cell death by cellular homeostasis or losing ions. In addition, it may denature the bacterial cell proteins along with various enzymes functions resulting in bacterial cell death [58,59].

The antiviral role of PCA against the virulent Influenza A virus subtype (H9N2) strain has been examined in mice by Ou et al. [60]. Mice were injected at a lethal dose of 0.2 mL (intra nasal) for H9N2 and three different concentrations including 40, 20 or 10 mg/kg PCA were considered as treatment groups for 7 days. The viral load (logEID50) was significantly lower in all the dosages of PCA than in the control. Besides, a higher survival rate of mice with PCA treated groups was recorded and the authors finally concluded that PCA could be an effective agent against bird flu infections and suggested to reinvestigate the potentialities of PCA in other diseases of poultry [60]. The lower viral load was due to the higher proportion of monocytes, CD8+ subset T cells, and a lower ratio of CD4+/CD8+ in the alveolar fluid of mice treated with PCA-rich phenolic polymers. Ou et al. [7] conducted an experiment in broilers challenged with virulent Infectious Bursal Disease Virus (IBDV) where PCA was applied as an effective agent to reduce the viral virulence. This study also reported that chickens were protected about 85% by PCA post-infection with IBDV [7]. Higher infectious bursal diseases showed viral clearance in the bursa of Fabricius of birds fed with 20 mg/kg or 40 mg/kg of PCA. In addition, higher lymphocyte proliferation and a higher CD4+/CD8+ ratio in a spleen sample of broilers fed with 20 mg/kg of PCA were also noted. The author highlighted that oral administration of 20 mg/kg PCA may induce nonspecific immune responses against viral diseases in broilers [7]. Infectious bursal disease viruses can replicate in the lymphocyte of bursa that can hamper the growth of B cells, resulting in apoptosis. Therefore, research was conducted to detect the pro-apoptosis role of PCA in IBD infection of broilers [53]. Birds were administered with 0.2 mL of 100 infectious dose-50 (ID50) of IBDV (oculo-nasal route) and PCA at 20 mg/kg was applied orally after 24 h of infection. The study reported that PCA could successfully alleviate the bursal pathological lesion at the early infection. PCA could influence the expressions of pro-apoptotic protein Bax and anti-apoptotic B-cell lymphoma 2 (Bcl-2); thus, gaur increase the progression of cell apoptosis resulting in inhibition of the infection of IBD. Furthermore, the authors also stated that PCA might have a role in regulating the PI3K/Akt and NF kappa B signal pathways in stimulating cell apoptosis post-IBDV infection [53]. Thus, PCA can be used as a potential antiviral agent in poultry production. The antiviral properties were further examined by Guo et al. [61]. In this study, specific pathogen-free chickens were offered PCA at 10, 20, and 40 mg/kg body weight to evaluate the humoral and cellular response of birds vaccinated with Newcastle disease virus (NDV). Antibody titers against NDV were significantly higher with PCA at 10–40 mg/kg on 7, 14, and 21 day of evaluation. Furthermore, PCA at 10–20 mg/kg could significantly decrease the viral load and shedding in proventriculus, compared with control birds or treated with Astragalus polysaccharides (positive control) which ensured the fact that PCA has an immune-enhancing role in farm animal production [61]. It was hypothesized that PCA might have a role in improving the humoral immune response that may lead to eradicating secondary viral infection in the experimental broilers. The authors also conducted a pathogen challenge trial where broilers were inoculated with NDV and PCA was considered as an agent for treatment. The survival rate was recorded about 50% in the challenge control group while the rate was higher—about 60 to 70% in PCA treated birds at 20 to 40 mg/kg, respectively, on day 14 [61]. Thus, collectively, the authors concluded that PCA can be an alternative antiviral agent that may improve the immunity of broilers and may enhance their health. The mechanism of antiviral activities of PCA is not clear yet in the literature. However, there may have numerous possibilities. It was hypothesized that PCA could induce immune regulation by developing the immune organs that could protect the host from infections. Alternately, PCA may have a role in improving the function of B cells, resulting in an increased production of antibody in response to microbial infections. In addition, the mechanism for the anti-viral role of PCA has been reported by decreasing the discharge of HBsAg and HBV DNA from HepG2 [10].

### 3.3. Role of Protocatechuic Acid on Lipid Metabolism, Immunity and Inflammation

The anti-atherosclerotic properties of PCA are well known because they can suppress the deposition of intracellular cholesterol. PCA-rich *Hibiscus sabdariffa* extract was proved as a good hypo-lipidaemic agent that could significantly decrease total cholesterol (TC), very low-density lipoprotein cholesterol (VLDL-C), low-density lipoprotein cholesterol (LDL-C), LDL-C:HDL-C risk ratio, in serum of experimental rats [62]. The authors claimed that the anti-atherogenic role of PCA was due to its anti-inflammatory properties. In a mouse model, PCA could prevent monocyte to TNF-𝛼 activated aortic endothelial cells, resulting in decreasing the adhesion of vascular cells and reducing the binding of NF-κB [62]. Lower plasma glucose with higher insulin levels was reported in mice fed PCA at 1%, 2%, and 4% [16]. In a recent study, high fat and fructose-rich diets were fed to male rats while PCA at 100 mg/kg was applied to the treatment group [63]. Rats in the treatment group showed lower triglyceride (TG) and total cholesterol (TC) along with the lower coronary risk index and atherogenic index in plasma. Then, the authors confirmed that PCA was a very effective agent against cardio-protective and hypo-lipidaemic properties. PCA at 0.003% (wt:wt) was fed to mice for 20 weeks and it was found to inhibit atherosclerosis development by controlling inflammatory responses [64], the molecular mechanism behind the upregulation of MERTK and inhibition of MAPK3/1 in macrophages. Oral administration of PCA (100 mg/kg BW) for 20 days could minimize the excessive level of hepatic marker enzymes (ALT, AST, ALP) and could decrease the serum lipid profiles (TC, TG, free fatty acids, phospholipids) than D-galactosamine-induced hepatotoxic rats [65]. The role of PCA on attenuating fatty acid-induced stenosis of mice was examined by Liu et al. [66]. PCA at 1%, 2% or 4% was added in diets of mice and found that PCA could significantly decrease the mRNA expression of lipogenic factors, decrease the deposition of hepatic 18:1 and 18:2 fatty acids, and could minimize the pro-inflammatory cytokines. It was also hypothesized that PCA could stimulate the flow of fecal sterols and decrease the absorption of dietary cholesterol resulting in a smaller deposition of lipid in plasma.

It was reported that the higher intake of rich fat diets causes hyperlipidemia in the blood and hepatic region, which leads to hepatotoxicity resulting in various metabolic disorders [67]. Metabolic disorders refer to the functional disorders of crude protein, fat, crude fiber, carbohydrate and other micronutrients in the host which creates hyperglycemia, hypertension, fatty liver diseases, abnormal growth and the development of different body tissues, etc. [19,68]. In farm animals, high-energy diets are usually applied with a view to producing more body weight and meat from commercial broilers or beef-producing animals. Thus, diseases related to metabolic disorders are very common in farm animals. Therefore, there lies a potential risk to human health through consuming high-fat food originating from these animals. PCA was very effective in restoring metabolic disorders in mice by reducing the body mass index, controlling blood glucose level, and insulin as well as restoring hormone levels for its antioxidant enzyme activities [69]. Based on different laboratory studies, Song et al. [19] highlighted that PCA could be a safe and potential therapeutic agent for controlling metabolic disorders.

PCA-rich phenolic compounds extracted from different natural herbs have showed immune-enhancing and anti-inflammatory properties that can be used as natural feed additives [17,70]. The mechanism may be due to the increased phagocytosis with the presence of PCA by the secretion of immunoglobulin and prohibiting the secretion of pro-inflammatory cytokines [13]. The anti-inflammatory properties of PCA have also been proven by lowering the discharge of interleukin (IL)-1 beta, tumor necrotic factor-alpha (TNF-a), and prostaglandin E2 in the brain tissue of experimental mice. The authors claimed that PCA might be an effective agent to mitigate the aging effects of mice by protecting the brain from inflammation and glycative damage [71]. In response to lung inflammatory properties, the concentration of neutrophil in the bronchial alveolar fluid was significantly higher while pro-inflammatory cytokines, namely interferon-c (IFN-c), IL-2, TNF-a and IL-6, were significantly decreased in the PCA-treated mice compared to the control, which ensured that PCA possesses anti-inflammatory properties [60]. In addition, PCA was found to reveal an inhibitory effect on nitric oxide (NO) production and TNF-a secretion in lipopolysaccharide interferon-c induced macrophages [60]. All anthocyanins including PCA showed angiotensin-converting enzyme (ACE)-inhibitory activities. PCA could decrease TG content in plasma, heart, and liver samples, and could lower the concentration of IL-6, TNF-𝛼 in the heart and kidney [16]. Thus the author claimed that PCA can be useful for lipid-lowering and anti-inflammatory effects. The basic mode of action of PCA on anti-inflammatory effects was via regulating NF-κB and MAPK activation, thereby decreasing TNF-𝛼 and IL-1𝛽, NO and PGE2 in RAW 264.7 cells and in vivo (mice model) [72]. Furthermore, the mode of action is correlated with decreasing the TLR-4 relaying response of the AKt, mTOR and NK-κB signaling pathways and the regulation of JNK and p38 MAPK activation [19]. Thus, PCA has the potential role to reduce the risk of inflammatory diseases like inflammatory bowel diseases, arthritis, colitis, cardiovascular diseases, etc. Moreover, the anti-inflammatory properties of PCA have been proven in different cell cultures by inhibiting monocyte infiltration via lowering CCR2 protein with mRNA expression and by reducing NF-κB activation [73,74]. LPS-induced sepsis mice were used as a model where PCA at 50 mg/kg was applied as the main treatment to detect the anti-inflammatory properties of PCA [75]. In this study, PCA exhibits the sepsis prevention properties by inhibiting the inflammatory cytokines (TNF-𝛼) and by lowering the concentration of ALT and nitrite/nitrate levels in plasma [75]. In a recent study, the protective role of PCA against inflammatory stress-induced human dermal fibroblasts was examined by Son et al. [2], while no cellular toxicity was reported under 100 µM with or without LPS. PCA could successfully mitigate the LPS-induced excessive ROS generation in human dermal fibroblast cells. In addition, PCA had a senescence-attenuating function by lowering senescent cells and by regulating the expression of *COL1A1* and the *MMP1* gene of type 1 collagen.

Existing studies on the anti-inflammatory and hypo-cholesteric properties of PCA are mostly based on laboratory animals and in vitro (cell culture) experiments [19]. Therefore, further studies focusing on the biological properties of PCA are extremely necessary to get more consistent data to improve the health status of farm animals.

### 3.4. Protocatechuic Acid on Intestinal Health Parameters

The gut system of mammals consists of massive and a complex microbial community. However, the microbes of the host maintain a symbiotic environment and directly or indirectly take part in the immune response [76]. Therefore, the relationship between the gut environment and the microbes is important to maintain for homeostasis of animals. A healthier gut system is very crucial to improving the utilization of nutrients for the proper growth and development of immune systems in animals [77]. Maintaining the health of an animal’s gut is possible by dietary manipulation. Therefore, there is a growing level of attention on the use of different dietary supplements that can alter the gut morphology, physiology, microbial diversity, etc., and provide a suitable gut environment for optimum growth and performance of farm animals. It was reported that different phenolic compounds or PCA, or their combination, played a vital role in improving the gut health of food animals. The mechanism behind the PCA-rich phenolic compounds could manipulate the gut microbial population, resulting in the suppression of pathogenic bacteria and improvement of the immune response [78].

Intestinal microbes can play a vital role in host metabolism. Numerous studies have noted that the major functions of gut microbes include the activation of the immune system [79], modulating the metabolism of lipids [80] and bile acids [81]. In specific terms, the gut microbiota can play an important part in the intestinal barrier function of animals. PCA has been reported as an effective agent as a counter to intestinal barrier dysfunction of mice [82]. Adding phenolic-rich oregano essential oil or acidifiers to animal diets could increase the amount of Lactobacilli and Bifidobacterium, resulting in higher lactic acid content in the intestine which could stimulate the expression of cytokines and up-regulate the expression of TGF-β and anti-inflammatory cytokines (interleukin-10, IL-10). In addition, it prevented the secretion of pro-inflammatory cytokines such as TNF-α and IL-6, and ensured a healthier gut system [83,84]. In swine models, the addition of polyphenolic-rich diets could increase the immunoglobulin G (Ig G) concentration in serum and could suppress pro-inflammatory genes in the intestine, which was hypothesized as beneficial to protect the intestinal surface from bacterial or oxidative damage and to maintain optimal intestinal function and gastrointestinal development [85,86].

Gastrointestinal pH is very important to secrete enzymes for optimum utilization of nutrients. For example, the pepsinogen enzyme could activate rapidly at a pH of 2 and the activation rate was slowed down with a higher pH of 4 [87]. Moreover, the optimum pH value for pepsin is 2–3.5, and the activity rate was decreased with the increased pH value [88]. Therefore, keeping the pH acidic in the gastrointestinal tract can increase the activity of pepsin. The lower pH of the chyme can stimulate the intestinal cell to secrete the pancreatic enzyme. In addition, a lower pH can also reduce the gastric emptying rate, resulting in the promotion of hydrolysis of the protein by pepsin [89]. The addition of different phenolic-rich compounds to the feed may reduce the pH of the gastric juice by lowering the pH of the diet. Some studies also documented that dietary inclusion of PCA-rich phenolic compounds or organic acids could significantly reduce the gastric pH [90,91]. In addition, a lower pH in the gastrointestinal tract could prevent the invasion and proliferation of harmful bacteria such as *E. coli* and *Salmonella* spp. in swine [92]. De Busser [93] reported that the addition of different acids to drinking water could significantly reduce the number of *E. coli* bacteria in pigs when the pH of their drinking water was at 4.0 or below. Dietary inclusion of PCA or organic acids could lower the pH in the stomach and was able to reduce the population of enterobacteria in the stomach of swine [94]. In a recent study, broilers supplemented with PCA-rich phenolic compounds (thymol) could decline *E. coli* in ileum [95]. The authors claimed that the lower number of *E. coli* might have an impact on increasing the nutrient’s absorption in the intestine by altering the epithelial cells to enhance the villus area of birds. In addition, Dong et al. [96] also reported that dietary inclusion of tea polyphenol could significantly increase the mucosal immunity by secretion of anti-inflammatory cytokines (IL-2, IL-10) in the small intestine and ensured good intestinal health of pigs. PCA also could improve intestinal immunity and alter the gut microbiota in broilers to favor a healthy gut in chickens [97]. However, data on the dietary supplementation with pure PCA on intestinal health-related parameters of other farm animals are still inadequate.

## 4. Application of Protocatechuic Acid in Farm Animal Production with Knowledge Gap

At present, there are few related studies available about the suggested dosages of protocatechuic acid (PCA) in animal diets. Therefore, it is necessary to reinvestigate the effects of PCA on improving the production performance and health status of animals. Chinese yellow feathered broiler diets were supplemented with two different dosages of PCA (300 mg/kg diets and 600 mg/kg, respectively), and found that a 300 mg/kg diet was significantly effective in improving the feed conversion ratio (FCR). However, average daily body weight gain and daily feed intake were not affected during days 1–52 of the study period [97]. Considering the meat quality parameters, the shear force value of breast muscle was significantly lower with the 300 mg/kg PCA supplemented group than the control birds. In addition, dietary supplementation of PCA at 300 mg/kg could enhance the meat color, redness value (a∗) and decrease the yellowness value (b∗) at 24 h postmortem compared to the meat from animals on control diets. However, this study did not find any significant differences in pH or drip loss among the treatment groups [97]. The higher redness value of meat is acceptable to consumers while higher yellowness or lightness values are associated with pale-colored meat [98]. The authors finally concluded that PCA could improve the meat quality of broilers. Mahfuz et al. [17] also reported that phenolic-rich compounds might have a role in improving the quality of meat. In another study, the body weight of broilers was significantly higher with PCA at 10 to 40 mg/kg oral dosages compared with control diets on day 30 or day 40 [61]. In addition, immune organ indices, for example, thymus indices and spleen indices, were significantly higher in broilers with a 40 mg/kg level of supplementation than in the untreated control group. The authors hypothesized that the oral application of PCA may induce the active cellular immune response resulting in good health and higher body weight gain in experimental broilers. PCA-rich phenolic extracts have been reported to enhance the growth performance of farm animals [13]. However, the reason for enhancing the growth performance by those phenolic compounds is related to the secretion of higher digestive enzymes and good gut health through reducing the harmful bacteria or by altering the antioxidant and anti-inflammatory properties of the gut surface [52,99]. It was also hypothesized that PCA-rich phenolic complexes made of herbs may increase the aroma and palatability of diets, resulting in higher feed intake (FI) and body weight gain of animals [13].

A lot of debates on farm animal studies with different phenolic-rich compounds were noted from various studies. For instance, higher ADG, with improved FCR in broilers fed with PCA-rich polyphenolic compounds at 300 to 600 mg/kg diets during days 0–42 of the study [100]. In addition, laying hens fed with phenolic-rich compounds originating from eucalyptus leaf extract at 0.5 to 0.12 g/kg diets could improve egg production, egg mass, egg shell thickness during weeks 35–44 of the study [101]. However, no significant differences in average daily weight gain (ADG) and FCR were noted in ducklings fed with oregano essential oil powder at 100 mg/kg diets for a trial period of 1–35 days [102]. Moreover, no significant differences in ADG, FI, and FCR were found in weaned piglets fed with polyphenolic-rich diets at a level of 5% for days 28–56 of the study [103]. Thus, it is also notable that all types of phenolic compounds are not necessarily important because a few of those compounds had a negative impact; even higher dosages may have no effects in several species [18]. For example, higher dosages of condensed tannins in poultry diets could inhibit protein and fat digestion due to the binding capacity of biliary salt or by deactivating the capacity of different digestive enzymes [104,105]. In addition, bean extract rich with proanthocyanidin was reported to decrease the activities of alpha-amylase, trypsin, and lipase enzymes in chickens [106]. Further phenolic compounds originating from grape seeds could also inhibit the absorption of different minerals such as Fe, Zn and Cu in human intestinal cells [107]. An interesting finding with condensed tannins was that they could precipitate protein digestion, resulting in a lower growth rate in pigs [108,109], whereas higher body weight was recorded in broilers fed with hydrolyzable tannins [110]. This action may be associated with different sources, the nature of compounds, dosages with extraction procedures, diets, and host species. Except for a few groups of the phenolics family (tannins, saponins, toxic alkaloids), the rest of the compounds, including PCA, are not usually reported as an anti-nutritive [111]. Therefore, future research on PCA in farm animal production is urgently needed.

## 5. Conclusions

The experimental knowledge about the role of protocatechuic acid (PCA) was reviewed in this study. Although many advantages have been reported on using the plant-derived antioxidant-enriched PCA for alleviating oxidative stress and inflammation, adequate data are still not available in the case of the farm animal study. PCA can be applied as an alternative for antibiotic growth promoters, because it can reduce the pH value of feed, improve intestinal health by improving the pancreas secretion, resulting in higher nutrient digestibility, enhance the growth of beneficial bacteria, and improve animal production performance. In addition, PCA has no residue, drug resistance or toxic effect, so it can be widely used in farm animal diets. Considering the biological value of PCA as well as the source of natural antioxidants, further studies are essential to examine the feasibility of protocatechuic acid on the production performance and health status of farm animals.

## Figures and Tables

**Figure 1 animals-12-00741-f001:**
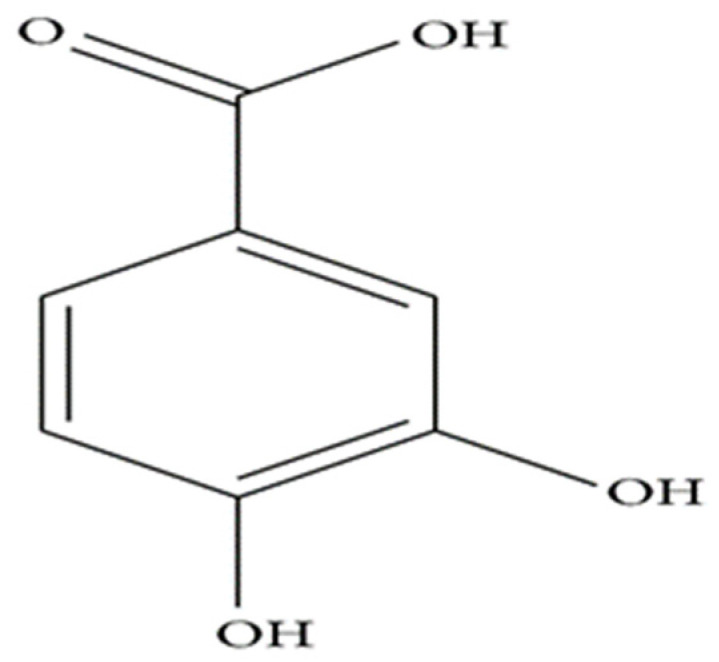
Chemical Structure of protocatechuic acid.

**Table 1 animals-12-00741-t001:** Common plant sources of protocatechuic acid.

Common Name	Scientific Name	Amount (mg/g)	Major Function	Ref.
Spinach tree leaf	*Cnidoscolus aconitifolius*	0.24	Antioxidant	Loarca-Pina et al. [24]
Danshen	*Salviae miltiorrhizae*	0.015–0.15	Antioxidant	Cao et al. [25]
Siberian Ginseng	*Acanthopanax senticosus*	1.43–5.56	Anti-inflammatory, antioxidant, hepato-protective	Huang et al. [26]
Grapes	*Vitis vinifera*	0.14–1.50 μg/g	Antioxidant	Liu et al. [27]
Almond	*Prunus amygdalus*	2.03–8.15 μg/mg	Antioxidant	Sang et al. [28]; Sivaci and Duman [29]
Wood speedwell	*Veronica montana*	15.7	Antimicrobial	Stojkovic et al. [6]
Queen Annes thistle	*Cirsium canum*	12–14	Antibacterial	Nazaruk [30]; Kozyra et al. [31]
Roselle	*Hibiscus sabdariffa*	0.09	Antibacterial/ antiviral	Chao and Yin [32]
Maidenhair tree leaf/ginkgo	*Ginkgo biloba*	21–44 (* GAE)	Antioxidant	Koczka et al. [33]
Palm tree/acai	*Euterpe oleracea*	0.04–2.14	Antioxidant and anti-inflammatory	Silveira and Godoy [34]; Pacheco-Palencia et al. [35]
Button mushroom	*Agaricus bisporus/arvensis*	5.1	Chemo-protective/immunomodulatory/Antioxidant	Aline et al. [36]; Dogan et al. [37]
Du-Zhong leaves	*Eucommia ulmoides*	17.2	Hepato-protective	Hung et al. [38]
Blue berry	*Vaccinium uliginosum*	0.30	Antioxidant, anticancer	Ryu et al. [39]
Brown berried juniper	*Juniperus oxycedrus*	0.73	Antioxidant	Taviano et al. [40]
Kadsura vine	*Kadsura longipedunculata*	0.79	Inhibiting the activity of alpha amylase	Cen et al. [41]
Holy basil or tulsi	*Ocimum sanctum*	0.101	Anticancer	Flegkas et al. [42]

* Total phenolic content, GAE, gallic acid equivalent.

## Data Availability

Not applicable.

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
