# Peer review of "Potential Role of Protocatechuic Acid as Natural Feed Additives in Farm Animal Production"

_animals, 2022, doi:10.3390/ani12060741_

Round 1

Reviewer 1 Report

Comments to the Authors of manuscript number: animals-1648067 entitled “Potential Role of Protocatechuic Acid as Natural Feed Additives in Farm Animal Production”.

The authors have presented a review about protocatechuic acid. They have presented possibility to use it as a nutritional supplement in animal breeding. It is especially interesting from the point of view of the replacement of the antibiotics which are prohibited in many countries.  Authors have presented the reason for which this prohibition is introduced, and showed numerous positive effects of the PCA use. It is very interesting revie worth to publish.

However, in general there is main problem that Authors do not use the uniform style in writing. When the abbreviation is introduced for the first time, use it please along the text; or there is many abbreviation without explanation. Authors have to remember that many different scientist could read this review, and not all can be specialist in the same field.

  1. L 20 – only “the review”
  2. L 22 – animal fodder
  3. L 23 – or animal breeding?
  4. L 28 – use the abbreviation
  5. L 28-29 – it is the Journal for animals not humans. Omit this sentence or rephrase
  6. L 30 – animal breeding
  7. L 31 – not animal diets but animal nutrition, especially that in L 35 there is spoken about farm animals
  8. L 32 – abbr.
  9. L 33 – animal nutrition
  10. L 64 – abbr.
  11. L 76- animal nutrition
  12. L 118 – use only abbr.
  13. L 121, 123 – the abbreviation should be explained
  14. L 124 – what about autophagy?
  15. L 136-137 – this sentence is not clear? What role?
  16. L 151- abbr. use
  17. L 176 – add infectious bursal disease virus
  18. L 181 – “IBD” explain please
  19. L 184 – what is ID50? It should be explained
  20. L 187 - B-cell lymphoma 2?
  21. L 194 – “NDV” explain abbr.
  22. L 214- use only abbr.
  23. L 217, 224 – TC – is it total cholesterol? Explain
  24. L 232 – FFA – explain
  25. L 253 – only abbr.
  26. L 258 – interleukin explained but TNF not
  27. L 263- IFN-c – explain it
  28. L 266 – explain abbr.
  29. L 328 – not everyone have to know that GI is gastrointestinal
  30. L 370 – add example of these indices
  31. L 388 ,390 – Average daily gain without abbr., then only abbr.

What is FI?

32. L 73 and 413 – what is this? 

Author Response

Reviewer 1

The authors have presented a review about protocatechuic acid. They have presented possibility to use it as a nutritional supplement in animal breeding. It is especially interesting from the point of view of the replacement of the antibiotics which are prohibited in many countries.  Authors have presented the reason for which this prohibition is introduced, and showed numerous positive effects of the PCA use. It is very interesting review worth to publish.

However, in general there is main problem that Authors do not use the uniform style in writing. When the abbreviation is introduced for the first time, use it please along the text; or there is many abbreviation without explanation. Authors have to remember that many different scientist could read this review, and not all can be specialist in the same field.

 Authors response: Dear Reviewer, thank you very much for reviewing our manuscript with good necessary comments, that’s all have been highly accepted by us. We have revised our manuscript accordingly. We have made all changes with color in text.  Below the point to point responses for your kind consideration.

  1. L 20 – only “the review”

Authors response: Thank you, we have revised it, Please check the L 20.

  1. L 22 – animal fodder

Authors response: Added, Please L 22.

  1. L 23 – or animal breeding?

Authors response : Added, Please L24

  1. L 28 – use the abbreviation

Authors response: Added, Please L28

  1. L 28-29 – it is the Journal for animals not humans. Omit this sentence or rephrase

Authors response: It has been rephrased as per advises.  Please L29.

  1. L 30 – animal breeding

Authors response: Added, Please L30.

  1. L 31 – not animal diets but animal nutrition, especially that in L 35 there is spoken about farm animals

Authors response: revised accordingly, Please L31.

  1. L 32 – abbr.

Authors response: added, Please L33.

  1. L 33 – animal nutrition

Authors response: revised accordingly, Please L33.

  1. L 64 – abbr.

Authors response: revised accordingly, Please L65.

  1. L 76- animal nutrition
  2. Authors response: revised accordingly, Please L77.
  3. L 118 – use only abbr.

Authors response: revised accordingly, Please L123.

  1. L 121, 123 – the abbreviation should be explained

Authors response: revised accordingly, Please L126-129.

  1. L 124 – what about autophagy?

Authors response: Dear Reviewer, It has similar meaning, Please check L129-131.

  1. L 136-137 – this sentence is not clear? What role?

Authors response: rewritten accordingly, Please L142-144.

  1. L 151- abbr. use

Authors response: revised accordingly, Please L154-159.

  1. L 176 – add infectious bursal disease virus

Authors response: revised accordingly, Please L183.

  1. L 181 – “IBD” explain please

Authors response: revised accordingly, Please L188

  1. L 184 – what is ID50? It should be explained

 Authors response: revised accordingly, Please L191.

  1. L 187 - B-cell lymphoma 2?

Authors response: Yes professor, you are correct. However, we have added full meaning accordingly, Please L195.

  1. L 194 – “NDV” explain abbr.

Authors response: added accordingly, Please L202.

  1. L 214- use only abbr.

Authors response: revised accordingly, Please L224.

  1. L 217, 224 – TC – is it total cholesterol? Explain

Authors response: revised accordingly, Please L226-228.

  1. L 232 – FFA – explain

Authors response: revised accordingly, Please L242.

  1. L 253 – only abbr.

Authors response: revised accordingly, Please L264.

  1. L 258 – interleukin explained but TNF not

Authors response: revised accordingly, Please L269-270.

  1. L 263- IFN-c – explain it

Authors response: revised accordingly, Please L274.

  1. L 266 – explain abbr.

Authors response: revised accordingly, Please L277-278.

  1. L 328 – not everyone have to know that GI is gastrointestinal

Authors response (AR): revised accordingly, Please L341.

  1. L 370 – add example of these indices

Authors response (AR): it was added here, Please L384.

  1. L 388 ,390 – Average daily gain without abbr., then only abbr.

What is FI?

Authors response (AR): revised accordingly, Please 394,402,

  1. L 73 and 413 – what is this? 

       Authors response (AR): we have added full meaning accordingly, Please L 74-75, 427.

Hope you will consider our revised submission and thanks a lot for your valuable time for us.

 Thank you.

Reviewer 2 Report

Dear author

Thanks for your submission in Animals journal. The concept of this review is fine and it has novelty in future research.  It’s a organized, well written submission and has important findings to the journal readers. Authors tried to highlighted the significant biological role of protocatechuic acid and explained the mechanism in the text. However, authors should consider some minor points as mentioned below  

Abstract:

L-   24-25: Providing safe food – industry; rewrite.

L25: delete-Natural

L29: replace as well as by >>and

Introduction

L40: delete the chemical structure () as you mentioned it under section 2.

L41-42: reported that the major metabolite of complex polyphenols (PCA), >>> reported that PCA, the major metabolite of complex polyphenols was….

L47: in vivo italic, Please check thoroughly in text.

L48:T1DM –full first.

L61:mg L-1??

L70-71: Instead, restriction on using antibiotics in animal feed creates demand for antibiotics alternatives in where??,rewrite it. Feed industry?

L76: Delete-as well as substitute for antibiotics

2.General Characteristics and Sources of Protocatechuic Acid

L93:g/Kg, please check the journal style and make similar in all units in the text.

3.1; L:120: check the unit pattern

3.3; L250: Based on the different…..

4.; L352: At present, there are no more related researches>>> At present, there are few related researches….

L406: Therefore the….

5.; L409: was reviewed..

L412: in farm animal study.

L419:of farm animals.

Author Response

Reviewer 2

Dear author

Thanks for your submission in Animals journal. The concept of this review is fine and it has novelty in future research.  It’s a organized, well written submission and has important findings to the journal readers. Authors tried to highlighted the significant biological role of protocatechuic acid and explained the mechanism in the text. However, authors should consider some minor points as mentioned below

Authors response: Dear Reviewer, thank you very much for reviewing our manuscript with good necessary comments, that’s all have been highly accepted by us. We have revised our manuscript accordingly. We have made all changes with color in text. Below the point to point responses for your kind consideration.  

Abstract:

L-   24-25: Providing safe food – industry; rewrite.

Authors response (AR): Thank you, we have rewritten this statement, please check the L24-26.

L25: delete-Natural

Authors response (AR): It was deleted from here, Please L26.

L29: replace as well as by >>and

Authors response (AR): revised accordingly, Please L29.

Introduction

L40: delete the chemical structure () as you mentioned it under section 2.

Authors response (AR): revised accordingly, Please L41.

L41-42: reported that the major metabolite of complex polyphenols (PCA), >>> reported that PCA, the major metabolite of complex polyphenols was….

Authors response (AR): revised accordingly, Please L42.

L47: in vivo italic, please check thoroughly in text.

Authors response (AR): revised accordingly in whole text, Please L47, 131 etc.

L48:T1DM –full first.

Authors response (AR): revised accordingly, Please L49.

L61:mg L-1??

Authors response (AR): revised accordingly in whole text. Please L62.

L70-71: Instead, restriction on using antibiotics in animal feed creates demand for antibiotics alternatives in where??, rewrite it. Feed industry?

Authors response (AR): revised accordingly, Please  L71-72.

L76: Delete-as well as substitute for antibiotics

Authors response (AR): it was deleted from here, Please L77.

2.General Characteristics and Sources of Protocatechuic Acid

L93:g/Kg, please check the journal style and make similar in all units in the text.

Authors response (AR): We have double checked the unit and ensured the similar pattern in text, thank you.

3.1; L:120: check the unit pattern

Authors response (AR): checked accordingly, Please L125.

3.3; L250: Based on the different…..

Authors response (AR): revised accordingly, Please  L261.

4.; L352: At present, there are no more related researches>>> At present, there are few related researches….

Authors response (AR): revised accordingly, Please  L366.

L406: Therefore the….

Authors response (AR): revised accordingly, Please  L420.

5.; L409: was reviewed..

Authors response (AR): revised accordingly, Please L423.

L412: in farm animal study.

Authors response (AR): revised accordingly, Please L426.

L419: of farm animals.

Authors response (AR): revised accordingly, Please L434.

Hope you will consider our revised submission and thanks a lot for your valuable time for us.

 Thank you.